# Caries Detection and Classification in Photographs Using an Artificial Intelligence-Based Model—An External Validation Study

**DOI:** 10.3390/diagnostics14202281

**Published:** 2024-10-14

**Authors:** Elisabeth Frenkel, Julia Neumayr, Julia Schwarzmaier, Andreas Kessler, Nour Ammar, Falk Schwendicke, Jan Kühnisch, Helena Dujic

**Affiliations:** 1Department of Conservative Dentistry and Periodontology, University Hospital, Ludwig-Maximilians University of Munich, 80336 Munich, Germany; lisafrenkel04@gmail.com (E.F.); julia.neumayr97@web.de (J.N.); julia.schwarzmaier@web.de (J.S.); andreas.kessler@med.uni-muenchen.de (A.K.); nour.ammar@outlook.de (N.A.); falk.schwendicke@med.uni-muenchen.de (F.S.); h.dujic@med.uni-muenchen.de (H.D.); 2Department of Prosthetic Dentistry, Faculty of Medicine, Center for Dental Medicine, Medical Center-University of Freiburg, University of Freiburg, 79106 Freiburg, Germany

**Keywords:** dental caries, diagnosis, validation study, artificial intelligence, deep learning

## Abstract

Objective: This ex vivo diagnostic study aimed to externally validate a freely accessible AI-based model for caries detection, classification, localisation and segmentation using an independent image dataset. It was hypothesised that there would be no difference in diagnostic performance compared to previously published internal validation data. Methods: For the independent dataset, 718 dental images representing different stages of carious (*n* = 535) and noncarious teeth (*n* = 183) were retrieved from the internet. All photographs were evaluated by the dental team (reference standard) and the AI-based model (test method). Diagnostic performance was statistically determined using cross-tabulations to calculate accuracy (ACC), sensitivity (SE), specificity (SP) and area under the curve (AUC). Results: An overall ACC of 92.0% was achieved for caries detection, with an ACC of 85.5–95.6%, SE of 42.9–93.3%, SP of 82.1–99.4% and AUC of 0.702–0.909 for the classification of caries. Furthermore, 97.0% of the cases were accurately localised. Fully and partially correct segmentation was achieved in 52.9% and 44.1% of the cases, respectively. Conclusions: The validated AI-based model showed promising diagnostic performance in detecting and classifying caries using an independent image dataset. Future studies are needed to investigate the validity, reliability and practicability of AI-based models using dental photographs from different image sources and/or patient groups.

## 1. Introduction

Dental caries is a prevalent, noncommunicable disease that requires appropriate diagnostics, oral health promotion, prevention and treatment measures [1,2]. By emphasizing non-invasive or minimally invasive caries management strategies, the importance of regular diagnostic evaluation is undebatable. Here, visual examination (VE) is the method of first choice [3,4,5,6,7], supported by supplementary X-ray-free or radiological diagnostic procedures when indicated [8,9]. In this context, all diagnostic procedures must have good validity, reliability and practicability. Notably, with respect to validity, false positive decisions should be avoided, as they can be potentially linked to overtreatment [10].

The importance of automated image analysis in medicine and dentistry has increased in recent years [11,12,13,14]. Automation can be achieved through the application of artificial intelligence (AI) methods. One data source for automated caries detection is digital photographs of teeth, which can be understood as the machine-readable equivalent of conventional visual examination. The few available data demonstrate promising accuracy of caries detection and classification in photographs using AI [13]. Notably, any such accuracy measures were generated using internal data, i.e., testing data stemmed from the same source as data used for training the AI-based algorithm. The associated risk of bias when only testing on such internal data has been demonstrated in several recent research projects [15,16,17,18,19,20].

While most studies on AI-based caries detection do not provide access to the developed algorithms, a recent model for caries detection, classification, localisation and segmentation developed by Felsch et al. [16] is freely accessible online as a web application. Unlike AI-based models for caries detection in X-rays, most of which are only available to dentists as paid software, this web application is the first to enable professionals to have clinical photographs of teeth automatically analysed by an AI-based model. Notably, such testing by others may be conducted independently from the workgroup publishing the data, i.e., using external test data. At present, the lack of external validations for AI-based models represents a significant knowledge gap in this field, highlighting the need for independent evaluations to ensure reliability and broader applicability. The aim of this study was to determine the external accuracy of this developed AI model for caries detection using an independent sample of images from the internet. It was hypothesised that the external validity would be identical to the internal validity published by Felsch et al. [16].

## 2. Materials and Methods

The present ex vivo diagnostic study is part of the “Caries detection with artificial intelligence” project at the Department of Conservative Dentistry and Periodontology, which received the approval of the Ethics Committee of the Medical Faculty of the Ludwig-Maximilians-Universität München (project number 020-798). The study was implemented in accordance with the guidelines of the Standards for Reporting of Diagnostic Accuracy Studies (STARD) Steering Committee [21]. In addition, the latest recommendations for standardizing the planning, implementation and publication processes of dental studies using AI methods were considered [22].

### 2.1. Collection of Dental Photographs from the Web

To externally validate the recently published AI-based model (http://demo.dental-ai.de; accessed 23 September 2024) [16], independent image data not involved in the development of the model were necessary. To fulfil this requirement, photographs of teeth with or without carious lesions were obtained from freely accessible internet sources. The following keywords were entered in a web browser (www.google.de) to search for photographs: “caries”, “primary dentition”, “permanent dentition”, “early childhood caries”, “beginning caries”, “noncavitated caries” and “cavity”. The inclusion criteria required photographs that were clear or exhibited minimal blur or distortion, had sufficient lighting, were without artefacts and had a minimum resolution of 72 pixels per inch. Exclusion criteria were images of teeth with direct/indirect dental restorations and tooth structure disorders, such as molar incisor hypomineralisation or fluorosis, as well as images showing rare dental diseases. The search and selection process took place over a period of two weeks and was carried out by the participating dentists, whereby uncertainties regarding the eligibility of inclusion and exclusion criteria were discussed in the working group. The identified photographs featured individual anterior and posterior teeth, with posterior teeth mainly captured from the occlusal view and anterior teeth from the vestibular view. Each photograph focused on single teeth rather than multiple teeth or a single arch. Regarding the previously defined inclusion criteria, a total of 287 and 431 photographs of deciduous and permanent teeth, respectively, were identified. The image set (*n* = 718) included 183 photographs of healthy teeth and 535 photographs of carious teeth; the latter group exhibited caries to varying degrees of severity. The photographs were saved in the given resolution and respective format to make them available for later image analysis.

### 2.2. Caries Detection and Classification by the Dental Workgroup (Reference Standard)

All included photographs (*n* = 718) were analysed, classified and agreed upon by the workgroup (EF, JN, JS, HD and JK). In the event of differing opinions, the image in question was discussed until a consensus was reached. The visual detection and classification of carious lesions were based on established criteria for caries detection [23,24,25,26,27]. In the first step, it was determined whether caries could be identified in the photograph or not (score 0—no caries). In the second step, differentiation was made regarding the degree of severity: 1—noncavitated caries (presence of opacities or discolouration indicating an established noncavitated lesion on the enamel surface), 2—greyish translucency/microcavity (greyish translucency of the enamel as a sign of undermining dentin involvement/caries-induced breakdown on the natural enamel surface), 3—cavitation (visible dentin involvement) and 4—destructed tooth (extensive cavity with near-total loss of the tooth crown). The categories were based on the established diagnostic classes from the available visual diagnostic systems [23,24,25,26,27] and the classes included in the AI-based model. In addition to the assessment of existing carious lesions by dentists, the photograph quality was evaluated and classified as either “acceptable” or “good”. This assessment included sharpness, resolution and exposure quality. When opinions in the workgroup diverged, the arguments for each classification were exchanged and discussed until a consensus was reached. The results of the dental evaluation provided a reference standard for the subsequent statistical exploration of the data. The reference standard was established before any AI-based image evaluation and can, therefore, be regarded as independent (Figure 1).

### 2.3. Caries Detection and Classification by the AI-Based Model (Test Method)

The above mentioned AI-based model be applied to detect caries and classify the clinical appearance as follows: 1—noncavitated caries, 2—greyish translucency/microcavity, 3—cavitation and 4—destructed tooth with nearly complete loss of the tooth crown [16]. For images showing teeth without caries, no pixel-level class was assigned by the AI-based model. Consequently, the class 0—no caries was documented for such images. The existing caries features were classified at a pixel level, allowing multiple caries classes to be identified simultaneously for each photograph. After accessing the abovementioned website, each photograph (*n* = 718) was uploaded separately, and the area of interest was narrowed, as required, using the crop tool, followed by AI-based image evaluation. The analysis was performed several weeks after the initial consensus on the reference standard. Each AI-based detection and classification result was verified within the workgroup (EF, JN, JS, HD and JK) to reach a group consensus.

### 2.4. Caries Localisation and Segmentation by the AI-Based Model

As part of the evaluation, the segments generated using the AI-based model were evaluated for correctness regarding localisation and segmentation (Figure 2). First, a comparison was made as to whether the localisation was accurately identified. For this purpose, at least one pixel from the marked segment had to lie within the corresponding carious lesion. If there was no pixel within the actual carious lesion, it was determined that the caries was incorrectly localised. In the second step, the quality of the segmentation was evaluated, and the predicted segment was compared with the present carious lesion. A distinction was made between a complete (approximately >90% match), partial (approximately <90% match) or absent match. The match could only be qualitatively estimated based on the photograph, as no quantitative image data were provided.

### 2.5. Data Management and Statistical Analysis

All photographic information and diagnostic decisions for the test method and reference standard were recorded in a digital datasheet created for this study. For this purpose, open-access data processing software was used (EpiData Manager and EpiData Entry Client, version v4.6.0.6, EpiData Association, Odense, Denmark, http://www.epidata.dk, accessed 23 September 2024). Before descriptive and explorative data analyses, data were exported and visualised using an Excel spreadsheet (Excel 2019, Microsoft, Redmond, WA, USA) and checked for validity. Using Python (version 3.8.5, http://www.python.org, accessed 23 September 2024), and the diagnostic performance of the test method relative to the reference standard was calculated. Specifically, the numbers of true positives (TPs), false positives (FPs), true negatives (TNs) and false negatives (FNs), the sensitivity (SE), the specificity (SP), the positive and negative predictive values (PPVs and NPVs, respectively), the diagnostic accuracy (ACC = (TN + TP)/(TN + TP + FN + FP)) and the area under the receiver operating characteristic (ROC) curve (AUC) were determined [28].

## 3. Results

The diagnostic performance of the AI-based image analysis (test method) was determined relative to the dental workgroup consensus (reference standard). In the first step, image-based caries detection was considered, resulting in a diagnostic accuracy of 92.0%. SE and SP were 92.0% and 91.8%, respectively (Table 1).

In the next step, all recorded carious lesions (*n* = 991) in the available photographs (*n* = 718) were considered. Cavitated (*n* = 326) and noncavitated lesions (*n* = 300) and caries-free teeth (*n* = 192) were the most frequent dental findings (Table 2). The overall agreement was 76.9%; there were 44 false positives (4.4%) and 185 false negatives (18.7%) (Table 2). The diagnostic performance of the test method across different lesion classes is shown in Table 3. Here, the overall diagnostic accuracy ranged from 85.5% (noncavitated caries) to 95.6% (destructed tooth). SE ranged between 42.9% (greyish translucency/microcavity) and 93.3% (noncavitated caries), and SP ranged between 82.1% (noncavitated caries) and 99.4% (destructed tooth). The AUC was between 0.702 (greyish translucency/microcavity) and 0.909 (no caries). The corresponding ROC curves are shown in Figure 2.

In addition, the diagnostic performance was determined considering the image quality of each digital photograph, which was classified as either “acceptable” or “good”. The calculated values can be taken from Table 4. We also evaluated the correctness of caries localisation and segmentation (Table 5). In principle, the AI-based method correctly predicted caries localisation in 755 cases (97.0%) but incorrectly in 23 cases (3.0%). Fully and partially correct segmentation was achieved for 52.9% and 44.0% of carious lesions, respectively, with 3.0% of the lesions being incorrectly segmented (Table 5).

## 4. Discussion

In the present ex vivo diagnostic study, a recently introduced AI-based model [16] was validated, and the results regarding its ability to automatically detect, classify, localise and segment carious lesions in digital photographs of teeth were determined. In summary, the AI-based model achieved an overall diagnostic accuracy of 92.0%, and the fraction of false-positive findings, which could be associated with overtreatment, was low (4.4%). In addition, most carious lesions were correctly localised by the test method (97.0%). The automatically drawn pixel segments were fully and partially correct in 52.9% and 44.1% of all cases, respectively.

The discussion of the documented external validity results for this new AI-based method is limited because 1) no other comparable AI-based diagnostic method is available thus far, and 2) most available diagnostic studies in the field of AI-dentistry on dental photographs involve model development and internal validation only [17,18,19,20,29,30]. Therefore, the aim of this study was to close this knowledge gap. A comparison between the internal [16] and external validity of the underlying AI-based model (Table 1, Table 2, Table 3 and Table 4, Figure 2) reveals higher accuracy in internal data (97.8%) than in external data (92.0%). Internal data refer to the AI-based model’s diagnostic performance on the dataset used during the development [16], which demonstrates the capability of the model to recognise patterns it has been trained on. In contrast, the performance of the AI-based model on external datasets reflects its ability to generalise to new and unseen cases. In view of the results obtained, the hypothesis that internal and external validity are identical had to be rejected. This decrease between internal and external validity has been reported in other dental [31] and medical study projects [32,33]. In a study by Fu et al. [31], the AUC for the detection of periapical lesions in 3D radiographs using an AI-based model decreased from 0.97 (internal validation) to 0.93–0.96 (external validation). The AUC values from the internal and external validations decreased from 0.949 to 0.843 in the AI-based detection of early childhood visual impairment [33]. Bora et al. [32] investigated the diagnostic performance of an AI-based model to detect diabetic retinopathy in adults; in their study, the AUC decreased from 0.81 (internal validation) to 0.71 (external validation) [32]. The abovementioned study data document the reduction in diagnostic performance from internal to external validation and thus emphasise the importance of external validation.

For the present study, clinical photographs of carious teeth of heterogenous quality were obtained from a variety of freely accessible internet sources; in contrast, model training was performed using well-standardised, high-quality, professionally captured photographs [16]. Therefore, the documented diagnostic performance (Table 1, Table 2, Table 3 and Table 4, Figure 2) supported the quality of AI-based caries detection, as all the photographs supplied to the model were unknown and nevertheless assessed with a high accuracy of 92.0%. Setting this order of magnitude from external validation in relation to the published data from other internal validation studies on caries detection generally showed relatively low or comparable values for diagnostic performance [18,19,20,29,30]. To date, only a few workgroups have focused on caries diagnostics using clinical photographs. Bottenberg et al. [34] detected caries in dental photographs and subsequently used histological examination of the extracted teeth as a reference standard. The AUC of 0.84 determined for dental diagnostics on photographs was comparable to our results, emphasising the general possibility of using photographs of teeth for diagnostic purposes [34].

A comparative analysis of the results for caries classification from the external validation (Table 2 and Table 3) and the internal validation data published by Felsch et al. [16] showed a slightly reduced diagnostic performance on our external validation data. This reduction in performance was evident in several caries groups (Table 2). For instance, in the noncavitated caries group, the ACC was 85.5%, the SE was 93.3%, and the SP was 82.1% (Table 3). Conversely, Felsch et al. [16] reported an ACC of 90.1%, SE of 88.4% and SP of 91.4%. For clinically more relevant cavitated lesions, the ACC was 87.3%, the SE was 71.2%, and the SP was 95.2, while Felsch et al. [16] reported an ACC of 95.9%, SE of 87.6% and SP of 97.8%. Similarly, Wang et al. [35] documented an ACC of 95.3% for the detection of healthy tooth surfaces versus tooth surfaces with white spot lesions. Notably, the authors integrated fluorescence data into the model, which could have increased the model performance [35]. Conversely, Thanh et al. [29] reported ACC values between 81.0% and 87.4% for the detection of healthy tooth surfaces and tooth surfaces with noncavitated and cavitated carious lesions. The internal model performance was lower than that of Felsch et al. [16] and Wang et al. [35].

As part of the external validation, the correctness of the localisation and segmentation of the AI-based model was evaluated. The recorded localisation was classified as correct in 97.0% of the cases. In contrast, segmentation was only rated as fully correct in just over half of the cases (52.9%), while in 44.1% of cases, the AI-based model achieved only partially correct segmentation of the existing carious lesions. However, these data require critical appraisal, as (1) they were subjective estimates by the participating dentists, (2) there are currently no recommendations for the qualitative and quantitative evaluation of segmented areas, (3) typical parameters, such as average precision or intersection over union, could not be determined, and (4) the selected cut-off of ~90% of the lesion area was quite strict. Therefore, these values should not be overinterpreted at this stage.

If the documented internal and external validation data are compared with the data from in vitro and/or in vivo studies on visual caries diagnostics, the data published to date for AI-based caries detection and classification on dental photographs indicate at least identical, if not improved, results. The meta-analysis by Macey et al. [7], in which the estimated summary SE and SP values were 0.86 (95% CI 0.80–0.90) and 0.77 (95% CI 0.72–0.82), respectively, should be mentioned here as a representative example. Another meta-analysis [6] confirmed this finding for caries detection on occlusal surfaces and revealed SE values of 0.70 (95% CI 0.59–0.80), SP values of 0.47 (95% CI 0.26–0.70) and AUC values of 0.70 based on the included clinical studies. On the one hand, these data show that AI methods can obviously generate accuracy similar to conventional VE methods. On the other hand, comparative clinical diagnostic studies, including meticulous VE and an independent, AI-based evaluation of the photographic tooth status, are lacking. This signals a knowledge gap that should be closed in future studies.

The present diagnostic study has strengths and limitations. Its novelty lies in being the first external validation study of the mentioned AI-based model for caries detection and classification using an independent sample of dental photographs with different caries classes as well as a range of image qualities. This aspect has to be understood as a unique feature of this study in comparison to previously published studies, which reported exclusively internal validation data [17,18,19,20,29,30]. External validation is a crucial evaluation step before recommending AI-based models for broader use in clinical practice. Therefore, this study addresses that knowledge gap and demonstrates the model’s ability to evaluate images from independent sources with promising diagnostic performance. Furthermore, this diagnostic study also provides information about caries localisation and segmentation (Table 5), which represent a new aspect of caries diagnostics. An additional novel aspect is that diagnostic accuracy was correlated with meaningful image quality (Table 4).

However, the following aspects should be discussed as limitations. To create an independent image database, photographs of teeth were deliberately taken from the internet. Although these photographs were of varying image quality and included a wide range of carious lesions and tooth surfaces, image size had only a limited influence on diagnostic quality (Table 4). This probably underlines the quality of the AI-based model, as it was also able to analyse photographs of only acceptable quality. Nonetheless, the possible bias in the selection of web images and the definition and application of inclusion and exclusion criteria, as well as the decision on the reference standard—which was always based on group consensus within the workgroup—should be noted. With respect to the use of dental photographs from external data sources, it was impossible for the workgroup to differentiate between real and AI-generated photographs. Furthermore, ground truth verification, which is primarily conducted by histological methods, was not available. This lack of information should be mentioned as another limitation. However, Bottenberg et al. [34] visually evaluated dental photographs for caries detection with acceptably high diagnostic accuracy relative to that of histological examination, thereby supporting the use of VE in cases where no other methods are applicable. Further, the ratio of available photographs with carious lesions (*n* = 535) to photographs showing healthy teeth (*n* = 183) could be a limitation. However, the workgroup made an effort to include all available photographs in the dataset, taking into account the inclusion and exclusion criteria. As this study mainly used images retrieved from the internet, the selected dataset showed qualitative loss compared to the dataset used to train and internally test the AI-based model [16]. To this extent, it can be discussed as a limitation, given that the described study design does not mimic the conditions of internal validation in its entirety. This aspect should be taken into account in future studies. Furthermore, the possibility of simultaneously conducting and documenting VE of the teeth under clinical conditions should also be considered, as this would allow further conclusions to be drawn regarding validity, reliability and generalisability. One challenge that emerged during the project was that the AI-based model provided multiple (pixel-wise) diagnoses per image. During most diagnostic studies, only one diagnostic decision is made for each tooth surface and method; thus, it is necessary to derive one diagnostic decision per image (Table 1) and consider the predicted segments regarding classification (Table 2, Table 3 and Table 4), localisation and segmentation (Table 5). In addition, the decision-making process was designed as a group consensus, as a completely independent evaluation of the segments against the dental reference standard was not possible. This is because the photographs in question had mostly multiple caries findings, which could not be blinded, potentially introducing verification bias, as noted here.

## 5. Conclusions

In this ex vivo diagnostic study, a recently introduced AI-based model was validated, showing promising diagnostic performance for caries detection, classification, localisation and segmentation using an external dataset of heterogeneous digital photographs of teeth. Diagnostic performance was slightly lower than in internal validation, reflecting challenges from varying image quality and lack of standardisation. With respect to these findings, future studies should investigate the validity, reliability and practicability of AI-based image analysis across different image sources and/or patient groups. In addition, standardised external validation protocols and the implementation of clinical assessments could improve the applicability of these models in different clinical settings.

## Figures and Tables

**Figure 1 diagnostics-14-02281-f001:**
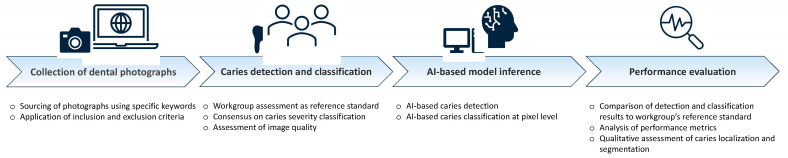
Workflow diagram illustrating the methodological steps.

**Figure 2 diagnostics-14-02281-f002:**
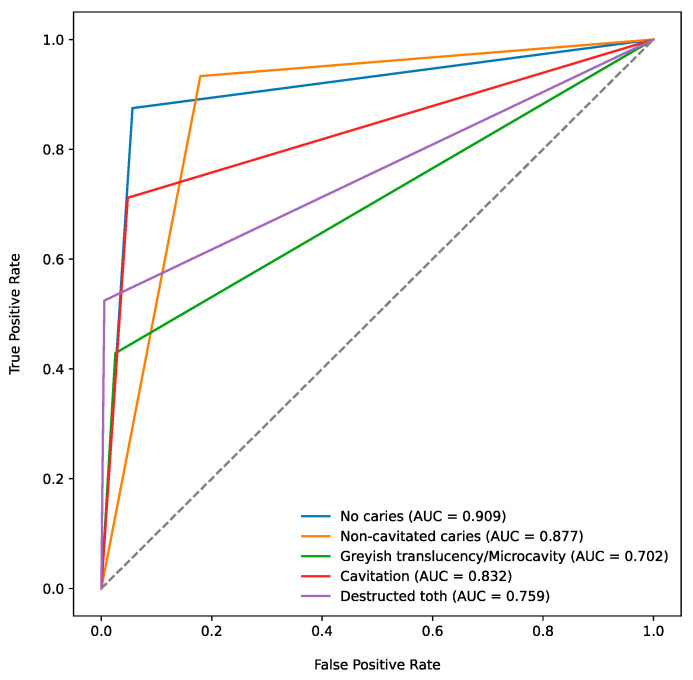
ROC curves and the corresponding AUCs for all caries classes.

**Table 1 diagnostics-14-02281-t001:** Cross-tabulation including diagnostic performance parameters for the AI-based image evaluation in relation to the visual consensus diagnosis by the dental work group (reference standard). In this analysis, only the ability for caries detection per image (*n* = 718) was considered.

Image-Related Caries Detection		Visual Evaluation (Reference Standard)			
		No Caries	Caries	∑	
AI-based evaluation (Test method)	No caries	168	43	211	NPV = 79.6%
	Caries	15	492	507	PPV = 97.0%
	∑	183	535	718	
		SP = 91.8%	SE = 92.0%		ACC = 92.0%

**Table 2 diagnostics-14-02281-t002:** Cross-tabulation of the AI-based image evaluation in relation to the visual consensus diagnosis by the dental work group (reference standard) for all caries diagnoses. This tabulation includes all diagnoses (*n* = 991) from all images (*n* = 718); multiple findings per image were possible.

		Visual Evaluation (Reference Standard)					
		No Caries	Noncavitated Caries	Greyish Translucency/ Microcavity	Cavitation	Destructed Tooth	∑
AI-based evaluation (Test method)	No caries	168	6	6	29	4	213
	Noncavitated caries	21	280	42	50	11	404
	Greyish translucency/Microcavity	1	9	39	13	0	62
	Cavitation	1	3	4	232	24	264
	Destructed tooth	1	2	0	2	43	48
	∑	192	300	91	326	82	991

**Table 3 diagnostics-14-02281-t003:** Summary of the diagnostic performance for all caries classes. The tabulation included all diagnoses (*n* = 991) from all images (*n* = 718).

	True Positives		True Negatives		False Positives		False Negatives		Diagnostic Performance (%)					
	*n*	%	*n*	%	*n*	%	*n*	%	ACC	SE	SP	PPV	NPV	AUC
No caries	168	17.0	754	76.1	45	4.5	24	2.4	93.0	87.5	94.4	78.9	96.9	0.909
Noncavitated caries	280	28.3	567	57.2	124	12.5	20	2.0	85.5	93.3	82.1	69.3	96.6	0.877
Greyish translucency/ Microcavity	39	3.9	877	88.5	23	2.3	52	5.2	92.4	42.9	97.4	62.9	94.4	0.702
Cavitation	232	23.4	633	63.9	32	3.2	94	9.5	87.3	71.2	95.2	87.9	87.1	0.832
Destructed tooth	43	4.3	904	91.2	5	0.5	39	3.9	95.6	52.4	99.4	89.6	95.9	0.759

**Table 4 diagnostics-14-02281-t004:** Summary of the diagnostic performance for all caries classes in relation to image quality. The tabulation includes all diagnoses (*n* = 991) from all images (*n* = 718).

	Image Quality	True Positives (*n*)	True Negatives (*n*)	False Positives (*n*)	False Negatives (*n*)	Diagnostic Performance (%)					
						ACC	SE	SP	PPV	NPV	AUC
No caries	Acceptable	95	277	19	17	91.2	84.8	93.6	83.3	94.2	0.892
	Good	73	477	26	7	94.3	91.3	94.8	73.7	98.6	0.930
Noncavitated caries	Acceptable	90	257	56	5	85.0	94.7	82.1	61.6	98.1	0.884
	Good	190	310	68	15	85.8	92.7	82.0	73.6	95.4	0.873
Greyish translucency/Microcavity	Acceptable	13	374	7	14	94.9	48.1	98.2	65.0	96.4	0.732
	Good	26	503	16	38	90.7	40.6	96.9	61.9	93.0	0.688
Cavitation	Acceptable	85	265	16	42	85.8	66.9	94.3	84.2	86.3	0.806
	Good	147	368	16	52	88.3	73.9	95.8	90.2	87.6	0.849
Destructed tooth	Acceptable	23	357	4	24	93.1	48.9	98.9	85.2	93.7	0.739
	Good	20	547	1	15	97.3	57.1	99.8	95.2	97.3	0.785

**Table 5 diagnostics-14-02281-t005:** Overview of the correctness of the AI-based caries localisation and segmentation. The analysis includes all caries diagnoses (*n* = 778).

	Localisation				Segmentation					
	Correct		Incorrect		Fully Correct		Partially Correct		Incorrect	
	N	%	N	%	N	%	N	%	N	%
Noncavitated caries	387	49.7	17	2.2	224	28.8	163	21.0	17	2.2
Greyish translucency/Microcavity	59	7.6	3	0.4	22	2.8	37	4.8	3	0.4
Cavitation	262	33.7	2	0.3	145	18.6	117	15.0	2	0.3
Destructed tooth	47	6.0	1	0.1	21	2.7	26	3.3	1	0.1
∑	755	97.0	23	3.0	412	52.9	343	44.1	23	3.0
	778 *				778 *					

* In total, 213 images were categorised as “no caries” by the AI-based model, for which no caries localisation and segmentation could be presented.

## Data Availability

The validated AI-based model is available as a web application and can be accessed at https://dental-ai.de (accessed on 23 September 2024). The data can be made available upon reasonable request.

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
