# Peer review of "Caries Detection and Classification in Photographs Using an Artificial Intelligence-Based Model—An External Validation Study"

_diagnostics, 2024, doi:10.3390/diagnostics14202281_

Round 1
Reviewer 1 Report (Previous Reviewer 1)
Comments and Suggestions for Authors
Dear Authors
all my suggestions have been addressed. Now the manuscript is suitable for publication.
Best regards
Reviewer 2 Report (Previous Reviewer 2)
Comments and Suggestions for Authors
Comments are addressed by the authors.
This manuscript is a resubmission of an earlier submission. The following is a list of the peer review reports and author responses from that submission.
Round 1
Reviewer 1 Report
Comments and Suggestions for Authors
Dear Authors
the topic of the manuscript is interesting but some changes are necessary before considering it for publication.
Here are my suggestions to improve it:
- Better formulate the abstract section describing the aim of the study.
- The introduction section resumes the existing knowledge regarding this topic but at the end of this section, Authors should underline the rationale of the study.
- In the central section, Authors should better clarify the inclusion and exclusion criteria.
- “In the first step, it was determined whether caries could be identified in the photograph. In the second step, a differentiation was made regarding the degree of severity: 0 – no caries, 1 – noncavitated caries, 2 – greyish translucency/microcavity, 3 – cavitation, and 4 – destructed tooth with nearly complete loss of the tooth crown”. If you did the two steps and in the first you divided the photos with cavities from those without cavities, what sense does it make to give the criterion 0 - no cavities a second time? Or what's the point of dividing the images without cavities if then in the images with cavities you classify the teeth as cavity free? Please specify.
- Furthermore, regarding point 2 and 3 of the caries classification, where you describe about microcavities and cavities from a photo, how can is it possible to understand if it is really decayed? I mean, you need the probe to know if there is a hole in the tooth. Please specify.
- A flow chart scheme is missing. It is adivisable to add it.
- Please specify the type of vision of the teeth in the photos. Occlusal? Lateral? 3/4? Etc... also are the shots homogeneous or the same for all cases or heterogeneous? And then they are photos of the whole mouth? Of a single arch? A hemiarch? A bunch of teeth? A single tooth?
- The discussion section appears well organized. Please add a specific sentence that clarifies the results obtained in the first part of the discussion.
- In the discussion it is stated that: “In summary, the overall diagnostic accuracy reached 92.0%, and the fraction of false-positive findings, which could be associated with overtreatment, was low (4.4%).” But how can is it possible to quantify true false positives and true false negatives from photos? I'm not referring to destructive cavities, but to little cavities (grade 1). Or how can you say that there is no cavity from a photo when in fact clinically, sometimes you just need to change your point of observation to view it?
- About "internal/external validity of the data to the AI", Plese better specify.
- The conclusion should reinforce in light of the discussions.
- Add the strenght and weakness of the study.
- References are updated.
Best regards
Comments on the Quality of English Language
English language is fine. Only minr spelling erros.
Best regards
Reviewer 2 Report
Comments and Suggestions for Authors
1. The title of the manuscript shall be revised. Include every step in the proposed method, not the name of every step.
2. The methodology is poorly written. A structured methodology is not included. From the manuscript, the methodology is solely done using an AI-based model (http://demo.dental-ai.de). If the author is not the developer of this AI-based model, the manuscript does not contribute to the extant knowledge by simply just presenting how the AI-based model is performed.
3. The novelty of the paper is not clear. Contributions are neglectable and are not suitable for this journal.
Round 2
Reviewer 1 Report
Comments and Suggestions for Authors
Dear Authors
All my requirements have been addressed. Now the manuscript has been improved and in my opinion it is suitable for publication.
Best regards
Comments on the Quality of English LanguageDear Editor
English language is fine. Only minor spelling errors.
Best regards
Reviewer 2 Report
Comments and Suggestions for Authors
The overall write-up is fine. However, the novelty and contribution are insufficient to be published in this journal. You may submit to some other journals (perhaps Q3 or Q4 journals). The comment herein mainly lies within the significance of the study, which in my opinion does not match the reputation of the journal. No additional results are added.
